# B1 Power Modification for Amide Proton Transfer Imaging in Parotid Glands: A Strategy for Image Quality Accommodation and Evaluation of Tumor Detection Feasibility

**DOI:** 10.3390/cancers16050888

**Published:** 2024-02-22

**Authors:** Xiaoqian Wu, Tong Su, Yu Chen, Zhentan Xu, Xiaoqi Wang, Geli Hu, Yunting Wang, Lun M. Wong, Zhuhua Zhang, Tao Zhang, Zhengyu Jin

**Affiliations:** 1Department of Radiology, Peking Union Medical College Hospital, Chinese Academy of Medical Sciences, Beijing 100730, China; 18075800553@163.com (X.W.); sutong_pumch@126.com (T.S.); xzt2013xzt@163.com (Z.X.); 15010147685@163.com (Z.Z.); jinzy@pumch.cn (Z.J.); 2Yangtze Delta Region Institute of Tsinghua University, Jiaxing 314006, China; peter@apodibot.com; 3Department of Clinical and Technical Support, Philips Healthcare, Beijing 100600, China; geli.hu@hotmail.com; 4Department of Stomatology, Peking Union Medical College Hospital, Chinese Academy of Medical Sciences, Beijing 100730, China; aquawyt@163.com; 5Department of Imaging and Interventional Radiology, Prince of Wales Hospital, The Chinese University of Hong Kong, Hong Kong 999077, China

**Keywords:** amide proton transfer, parotid tumor, image quality, magnetic resonance imaging

## Abstract

**Simple Summary:**

Amide proton transfer weighted (APTw) imaging is a contrast-free molecular imaging method based on the chemical exchange saturation transfer (CEST) technique, initially applied and explored in brain cancers. Previous studies of APTw imaging in the head and neck area have applied APTw protocols in the brain, and hyperintensity artifacts remain a problem to solve. A total of 32 lesions and 30 parotid glands were involved in this research to address the effect of B1 power modification on improving APTw imaging quality in parotid tumor identification, aiming to minimize hyperintensity artifacts. We found that hyperintensity artifacts declined with B1 power decreasing, and combinations of different APTw sequences could improve tumor detection feasibility compared to one APTw sequence. Our findings in this research could give an insight into APTw imaging quality improvement in the head and neck area, which might help the noninvasive diagnosis of parotid tumors in the future.

**Abstract:**

Background: In the application of APTw protocols for evaluating tumors and parotid glands, inhomogeneity and hyperintensity artifacts have remained an obstacle. This study aimed to improve APTw imaging quality and evaluate the feasibility of difference B1 values to detect parotid tumors. Methods: A total of 31 patients received three APTw sequences to acquire 32 lesions and 30 parotid glands (one patient had lesions on both sides). Patients received T2WI and 3D turbo-spin-echo (TSE) APTw imaging on a 3.0 T scanner for three sequences (B1 = 2 μT, 1 μT, and 0.7 μT in APTw 1, 2, and 3, respectively). APTw image quality was evaluated using four-point Likert scales in terms of integrity and hyperintensity artifacts. Image quality was compared between the three sequences. An evaluable group and a trustable group were obtained for APTmean value comparison. Results: Tumors in both APT2 and APT3 had fewer hyperintensity artifacts than in APT1. With B1 values decreasing, tumors had less integrity in APTw imaging. APTmean values of tumors were higher than parotid glands in traditional APT1 sequence though not significant, while the APTmean subtraction value was significantly different. Conclusions: Applying a lower B1 value could remove hyperintensity but could also compromise its integrity. Combing different APTw sequences might increase the feasibility of tumor detection.

## 1. Introduction

The parotid gland (PG) is the most common site of salivary tumors anatomically [1]. Characterization and definitive diagnosis of parotid tumors (PTs) are required before therapy arrangement. Fine-needle aspiration has been extensively applied clinically nowadays; however, its sensitivity is limited by parotid tumor heterogeneity and insufficient biopsy coverage [2,3]. As a complement, modern imaging can provide an overall insight into tumor characterization, including ultrasonography, computed tomography, and especially magnetic resonance imaging (MRI) with its dynamic and advanced sequences [4]. MRI is generated from a strong magnetic field (B0) and pulses of radiofrequency energy (B1) by a coil, free from ionizing radiation [5]. Conventional MRI enables precise evaluation of tumor size, location, border, growth pattern, and signal intensity on T1WI and T2WI [6]. Application of functional MRI in PT evaluation, such as diffusion weighted imaging (DWI) and dynamic contrast enhanced (DCE) MRI, reflect tissue characterizations of diffusion restriction, contrast enhancement, and washout to differentiate benign and malignant lesions, nonetheless with still conflicting results and measured value overlaps [7,8]. Therefore, advanced MRI techniques have been explored for qualitative diagnosis of parotid tumors [4].

Amide proton transfer (APT) imaging is a contrast-free molecular imaging method based on the chemical exchange saturation transfer (CEST) technique, in which amide protons are saturated by a selective saturation radiofrequency (RF) pulse at 3.5 ppm (parts per million) and the saturation is continuously transferred to the bulk water by chemical exchange, reducing the water magnetization and signals [9,10]. The saturation build-up of water molecules amplifies signal reduction, enabling detection of free proteins and peptides at low concentrations [11]. The CEST effect is usually small and is sensitive to B0 and B1 fields and saturation pulse power. APT application for tumor detection and characterization was first explored and is best understood in the brain [12,13]. Until now, more and more APT studies have been reported in breast [14,15,16], cervix [17,18], prostate [19,20,21], and chest tumors [22,23], demonstrating its feasibility for tumor detection and characterization. However, APT weighted (APTw) imaging studies in head and neck tumors face obstacles considering motion effects such as breathing and swallowing, although the parotid glands are thought to be less influenced by such motion disturbance [24].

Based on previous studies of APTw imaging of brain tumors, a fast 3D acquisition technique, integrated with a feasible, optimized RF saturation scheme and an effective lipid suppression method, is recommended and TSE has been recommended among the candidate readout sequences [25]. In most APTw studies of parotid cancers, the APT protocol recommended for brain cancers (B1 = 2 μT, saturation time = 0.8~2 s) has been applied [13,24,26,27]. However, although this power and saturation time guarantee homogeneity in most normal brain areas, when applied to the parotid region, inhomogeneity and hyperintensity artifacts in PG and PT remain a hindrance [28,29]. APTw studies in the brain have demonstrated that lower B1 could lessen hyperintensity artifacts [30]. We expect that hyperintensity artifacts in PGs and PTs could also be minimized and tumor detection efficacy could be improved by B1 decreasing.

Therefore, in this study we investigated APTw image quality improvement by B1 power modification and evaluated the feasibility of different APTw sequences to detect parotid tumors.

## 2. Materials and Methods

### 2.1. Prospectively Enrolled Participants

Inclusion criteria were as follows: (1) suspected diagnosis of PT clinically or by ultrasound, and (2) MRI examination plan. Exclusion criteria included severe image degradation which would impair further analysis caused by motion or dental artifacts. From October 2019 to December 2021, 35 consecutive patients meeting the inclusion criteria were enrolled in this study. Among them, four patients were excluded due to severe motion (*n* = 2) and dental artifacts (*n* = 2) that caused degradation of the anatomical planning images. Ultimately, 31 patients underwent 3D TSE APTw MRI on a 3T MRI scanner (Ingenia CX, Philips Healthcare, Amsterdam, The Netherlands) equipped with dual radiofrequency (RF) transmit coils and a 32-channel head-coil for image acquisition.

### 2.2. MRI

In this study, all patients underwent MR imaging using a 3.0T MRI system (Ingenia CX, Philips Healthcare, Amsterdam, The Netherlands) equipped with a 32-channel phased-array radiofrequency receiver coil. The scan sequences included axial T1w-TSE, axial T2w-TSE, and three APTw imaging protocols with different saturation pulse amplitudes: 2 μT, 1 μT, and 0.7 μT. The parameters were shown in Table 1. APTw images were reconstructed using the integrated software available in the MR control console [31].

### 2.3. Image Analysis

MR images and data were transferred to a workstation (Intellispace Portal; v. 10.1.0.64190; Philips Healthcare) for further analysis. Two independent reviewers, one radiologist (YC) and one radiology scientist (TS) with 24 and 10 years of head and neck experience, respectively, evaluated all MR images. Additionally, a third senior radiologist (ZJ) with 35 years of head and neck experience finally determined the score when there was inconsistency between the two readers. The three readers were blinded to all patients’ clinical and pathological data. Both the PTs and the PGs were evaluated on APTw images independently. The patients’ gender, age, and the largest lesion diameter in axial images were also recorded.

### 2.4. Qualitative Analysis

A qualitative analysis was performed to assess the image quality of APTw imaging, following the scoring strategy described in a previous study [28]. An area whose APTw value < −5% is defined as integrity loss, and one whose APTw value > 5% is defined as a hyperintensity artifact, both of which impair imaging quality. Two radiologists drew first regions of interest (ROI1) on the APTw images by manually outlining the entire PT or PG based on T2W images. The integrity and presence of hyperintensity artifacts within the ROI1 were evaluated using a 4-point Likert scale (4 = excellent, 3 = good, 2 = moderate, 1 = poor), shown in Table 2. PTs or PGs with score 1 or 2 in APTw images were excluded. The two radiologists independently scored the integrity and hyperintensity artifact of PTs and PGs, and the consistency between their assessments was analyzed. In cases where there were initially inconsistent scores, a third senior radiologist reviewed the images and made the final decision.

### 2.5. Quantitative Analysis

The quantitative analysis in this study involved the following steps: (1) Comparison of image quality: Scores for PTs and PGs were compared among three APTw sequences using the same 4-point Likert scale. (2) APTmean and subtraction values: A second ROI (ROI2) was drawn within ROI1, measuring APTmean values for PTs and PGs in each APTw sequence (using saturation B1rms of 2, 1, and 0.7 μT). The principle of drawing ROI2 is to maintain the largest area of normal gland or lesion while deleting hyperintensity areas, trying to obtain the real APTw value. APTmean and subtraction values were compared between PTs and PGs in the evaluable and trustable groups.

### 2.6. Statistical Analysis

The statistical analysis in this study was performed using IBM SPSS software, version 19.0 (SPSS Inc., Chicago, IL, USA). The differences of integrity and hyperintensity artifacts scores between tumor lesions and parotid glands were compared using the Mann–Whitney U test. The Wilcoxon paired signed-rank test was used to compare the difference of integrity and hyperintensity artifact scores between the three APTw sequences. A nonparametric test was used for comparing two related groups. The kappa coefficient was calculated to examine the correlation of integrity scores and hyperintensity artifact scores between the two radiologists. The Shapiro–Wilk test was performed to evaluate the normality of distribution of APTmean values. The paired sample *t*-test was used to compare the difference in APTmean values between the three sequences. APTmean values were presented as the mean ± standard deviation. All statistical tests were two-sided, and *p* ≤ 0.05 was considered statistically significant.

## 3. Results

### 3.1. Patients’ Characteristics

Among the 31 patients who underwent APTw examination, there were 21 men and 10 women with an average age of 57.74 ± 17.78 years (range, 15–92 years). One patient had lesions in both left and right sides of the parotid gland, and the normal parotid tissue was too small to evaluate, resulting in a total of 32 lesions and 30 parotid glands being included in the analysis. The average diameter of the lesions was 21.29 ± 8.28 mm (range: 9.1–52.6 mm). Out of the 31 patients, 22 received surgery and had pathological results available. Patients’ characteristics and pathological results are shown in Table 3.

### 3.2. Qualitative Analysis

In the images of the three sequences, four PTs and two PGs with an integrity score of 1 were excluded from the analysis to obtain an evaluable group for artifacts evaluation. In the evaluable group, which was consisted of 28 PTs and 28 PGs, hyperintensity was evaluated. Nine PTs and 11 PGs with a hyperintensity score of 1 or 2 were excluded to ensure trustable APTw value measurements. Consequently, 19 lesions and 17 parotid glands were enrolled in the trustable group for APTw value measurement. The patient selection flowchart is shown in Figure 1.

For qualitative analysis, the region of interest (ROI1) of each PG and PT was manually delineated based on the T2W images by the two radiologists. ROI2 was drawn in such a way as to maintain the most parotid gland or lesion area while excluding most of the hyperintensity artifacts (Figure 2).

Integrity and hyperintensity artifact scores were assessed independently by two radiologists for PTs and PGs in ROI1. As shown in Table 4, the integrity and hyperintensity artifact scores made by the two radiologists had great consistency. The scores used for subsequent quality comparison between sequences were based on the final scores determined by the third senior radiologist.

The integrity and hyperintensity artifact scores were compared between PTs and PGs for each APT sequence. Statistical significance was determined using appropriate statistical tests, and *p*-values are shown. No significant difference indicates *p* > 0.05. The results of qualitative analysis showed that there was no significant difference in integrity between PTs and PGs in APT 1 and APT 2 (*p* = 0.082 and 0.723, respectively), while PGs had better integrity than PTs in APT 3 (*p* < 0.05). After excluding PTs and PGs with integrity score = 1, the hyperintensity artifact analysis revealed that PTs had better image quality of hyperintensity compared to PGs in APT sequences 2 and 3 (*p* < 0.05), while there was no significant difference in APT sequence 1 (*p* = 0.669). Further details and a case example can be seen in Table 5 and Figure 3, respectively.

The results of the comparison among the three APTw sequences (APT1, APT2, and APT3) showed that PTs had better integrity in APT 1 compared to APT 2, and APT 2 had better integrity compared to APT 3. However, there was no significant difference in integrity between the three sequences for PGs. In terms of hyperintensity artifacts, lesions in both APT 2 and APT3 had less hyperintensity artifacts compared to APT1 (*p* < 0.05), while there was no significant difference in hyperintensity artifacts between APT 2 and APT 3 for PTs (*p* = 0.157). Similarly, PTs in APT 2 and APT 3 also had less hyperintensity artifacts compared to APT 1 (*p* < 0.05), and there was no significant difference between APT 2 and APT 3 (*p* = 0.083). Further details can be seen in Table 6.

### 3.3. Quantitative Analysis

In the evaluable group, there was a significant difference in APTmean between PTs and PGs in sequence 2 and sequence 3 (0.60% ± 0.75% vs. 1.46% ± 1.82% and 0.06% ± 0.83% vs. 0.88% ± 1.85%, respectively), while there was no significant difference in APTmean between PTs and PGs in sequence 1 (3.18% ± 2.74% vs. 2.21% ± 2.12%, *p* = 0.144). PTs had significantly higher APT1mean–APT2mean and APT1mean–APT3mean values compared to PGs (2.59% ± 2.48% vs. 0.75% ± 1.20%; 3.13% ± 2.64% vs. 1.33% ± 1.48%).

In the trustable group, there was no significant difference in APTmean between PTs and PGs in all three sequences. However, PTs had significant higher APT1mean–APT2mean and APT1mean–APT3mean values compared to PGs (1.49% ± 1.54% vs. 0.20% ± 0.51%; 1.91% ± 1.88% vs. 0.51% ± 0.56%). The quantitative analysis results of the three sequences are shown in Table 7.

## 4. Discussion

APTw imaging is a contrast-free molecular imaging method based on the chemical exchange saturation transfer (CEST) technique, initially applied and explored in brain tumors. Previous studies of APTw imaging in head and neck area apply APTw protocols in the brain, and hyperintensity artifacts remain a problem to solve. This study aimed to investigate the effect of B1 power modification in 3D TSE APTw for parotid gland tumor detection, with the target to minimize hyperintensity artifact in APTw images. A total of 32 lesions and 30 parotid glands were involved in this research to address effect of B1 power modification. The study found that B1 power decreasing resulted in decreased hyperintensity artifacts with the expense of losing image integrity. The comparison of APTmean values between lesions and parotid glands revealed that the subtraction value of APTmean between two sequences might be a stronger parameter than APTmean value in one single sequence, indicating that combining APTw sequences with different B1 powers could potentially improve parotid tumor detection. Our findings in this research could give an insight into APTw imaging quality improvement in the head and neck area, which might help the noninvasive diagnosis of parotid tumors in the future.

In terms of data analysis of this study, the choices of statistical tests used and their suitability for the type of data and research questions posed are discussed thoroughly here. A 4-point Likert scale was used to quantify the image quality of APT. The Mann–Whitney U test was used to compare the differences of integrity and hyperintensity artifacts scores between tumor lesions and parotid glands, because its application is to compare the differences between two independent samples when the sample distributions are not normally distributed and the sample sizes are small (*n* < 30). The Wilcoxon paired signed-rank test was used to compare the difference of integrity and hyperintensity artifact scores between the three APTw sequences for the same reason. As for APTw values, the Shapiro–Wilk test was performed firstly to evaluate the normality of distribution, and proved its normality. Therefore, the paired sample *t*-test was used to compare the difference in APTmean values between the three sequences.

APTw MRI images in parotid tumors are mainly studied to differentiate malignant and benign tumors. Bae et al. studied APTw imaging on 23 benign and 15 malignant parotid tumors, finding mean APTw values had better diagnostic performance compared with maximum and median APTw values [32]. Kamitani et al. compared mean APTw values between 21 benign and 12 malignant parotid tumors, and found significantly higher APTw values of malignant tumors (2.99 ± 0.99% vs. 2.23% ± 0.80%, *p* = 0.01) [24]. Law et al. also addressed similar findings of APTw values between malignant and benign parotid tumors, and also found that adding APTw value to ADC could increase the area under the curve from 0.87 to 0.96. These studies on differentiation between malignant and benign parotid tumors, however, depended on other MRI images to recognize parotid tumors in APTw images. It remains a question for parotid tumor detection in APTw imaging whether the APTw value between parotid tumors and normal parotid glands is significantly different. Our previous study showed differences in APTw values between lesions and parotid glands only in trustable samples after excluding images with hyperintensity artifacts, suggesting that hyperintensity artifacts are a notable interference factor in the application of APTw MRI for parotid lesion detection [28], which led to this study to decrease the interference of hyperintensity artifacts, and to improve image quality of APTw imaging for parotid tumor detection.

CEST imaging is sensitive to B0 and B1 field inhomogeneity, which can induce artifacts [33,34]. In this study, the effect of B0 inhomogeneity on APTw values was minimized by z-spectrum correction based on the B0 field map. Other studies have shown that APTw values increase with B1 power, while CEST contrast is not sensitive to B1 inhomogeneity [35], indicating the potential of using weaker RF to decrease hyperintensity artifacts without attenuating APTw contrast. When B1 = 0.5 μT was applied in our pre-experiment, there were large areas of signal loss which impaired tumor detection, so we chose a B1 gradient of 2, 1, 0.7 μT. In this study, it was found that with decreasing B1 power, the artifacts of both PTs and PGs declined, although with integrity loss of lesions.

Theoretically, APTmean value would be higher in PT compared to PG due to the increased content of mobile protein and peptides, resulting in an increased amide proton transfer rate in tumors [12]. However, previous studies have shown controversial results regarding APT values between parotid lesions and parotid glands. Yuan et al. found the mean APTw value of 12 normal parotid glands was 7.62%, while the APTw value of parotid pleomorphic adenoma was 1.18% [27]. Chen et al. found the difference between parotid tumors and parotid glands is significant in the trustable group (1.99% ± 1.18% vs. 1.03% ± 1.09%, *p* = 0.018) [28]. This study found different results depending on the B1 power used: APTw values of PTs were higher than PGs in strong B1 power (B1 = 2 μT), although not significant, but lower in weaker B1 power. That significant differences were found in evaluable groups, while not in trustable groups (see in Table 7), might be explained as the limitation of the sample size of trustable groups. In addition, the difference of APT1 value between PTs and PGs of the evaluable group was found not significant, possibly because the distinction of APTw values was small and a relatively larger sample size was required. Moreover, it was an interesting finding that APTmean of lesions might decease more than normal parotid glands with decreasing B1 power. Therefore, the difference in APTmean subtraction values between PTs and PGs was compared, revealing a significant difference as expected.

The positive results of the comparison between sequences might be due to several reasons. First, the signals collected by APT sequences from parotid glands and tumors can be easily corrupted by slight body motions, such as swallowing, etc. [33]. The subtraction of APT signals collected from two RF powers might have the potential to reduce motion impact. Second, the signals obtained from APT sequences consist of APT signal, nuclear over Hauser enhancement (NOE) effect and conventional magnetization transfer (MT) signal, which have different compositions in normal and tumor tissues and have different trends of signal strength with B1 change [13]. The subtraction of signals from two RF powers could possibly weight the key factors that matter most, leading to better results compared to using a single RF power. Thirdly, B1 power might have a different extent of impact for different molecules, which could be presented in the APTw subtraction values in different B1 powers between lesions and parotid glands.

The study has several limitations that should be acknowledged. First, the limited number of patients included in the study may affect the generalizability of the findings. Patient selection criteria and exclusion of unreliable images may have led to a smaller sample size, which could decrease the confidence level of the data. According to our published pre-experiment [28], the difference between parotid tumors and parotid glands is significant in the trustable group (1.99% ± 1.18% vs. 1.03% ± 1.09%, *p* = 0.018), when α = 0.05, β = 0.2, μ1, μ2, σ1, and σ2 are set as listed above, N1 and N2 are calculated as 23 and 23 to obtain a significant result, respectively (PASS 2021, v21.0.3). Therefore, 23 parotid tumors and 23 parotid glands in a trustable group are required to find a significant difference. In this study, after evaluation and selection, only 19 parotid tumors and 17 parotid glands were left in the trustable group in the analysis. Further studies with larger sample sizes are needed to validate the findings. Second, the modification of B1 power in this study was performed using only three gradients, which may not have captured the optimal balance between maintaining integrity and decreasing hyperintensity artifacts. More precise parameter modifications and optimization of B1 power may be necessary to achieve better results in future studies. Third, while the study provided some possible explanations for the APTmean subtraction values between sequences, the underlying theoretical representation remains to be discovered. The exact mechanisms and molecular basis for the observed differences in APTw values between different B1 powers and their impact on parotid tumor detection are not fully understood and require further investigation.

In conclusion, despite the findings of this study, there are limitations that should be taken into consideration, including the small sample size, limited parameter modifications, and the need for further research to better understand the underlying mechanisms. Future studies with (1) larger sample sizes: as calculated above, at least 23 parotid tumors and 23 parotid glands after selection in trustable group; (2) more precise parameter modifications: the B1 power gradient decrease and specification especially; and (3) in-depth investigations into the molecular basis of APTw MRI findings are recommended to validate and expand on the results of this study.

## 5. Conclusions

The discoveries of this study suggest that modification of B1 power might improve APTw image quality in parotid tumor identification by reducing hyperintensity artifacts, although at the expense of losing integrity. Combining different APTw sequences may enhance the efficacy of parotid tumor detection, potentially overcoming the limitations of individual sequences. Additionally, the study highlights the need for precise parameter modifications in APTw MRI for further quality improvement, particularly in the challenging head and neck area where hyperintensity artifacts can be prominent. The findings of this study contribute to the understanding of the potential benefits and limitations of B1 power modification in APTw MRI for parotid tumor identification. However, it is important to acknowledge the limitations of the study, including the small sample size and the need for further research to better understand the underlying mechanisms of APTw MRI findings. Future studies with larger sample sizes, more precise parameter modifications, and comprehensive evaluation of different APTw sequences are warranted to validate and expand on the results of this study, to ultimately improve the clinical utility of APTw MRI in the detection and characterization of parotid tumors.

## Figures and Tables

**Figure 1 cancers-16-00888-f001:**
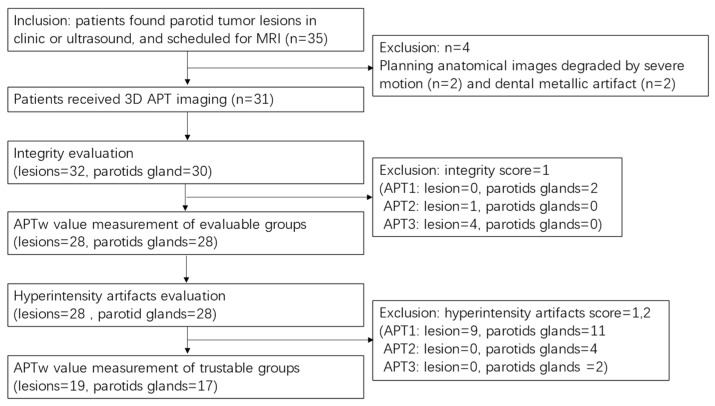
Patient selection and APT evaluation flowchart. APT, amide proton transfer.

**Figure 2 cancers-16-00888-f002:**
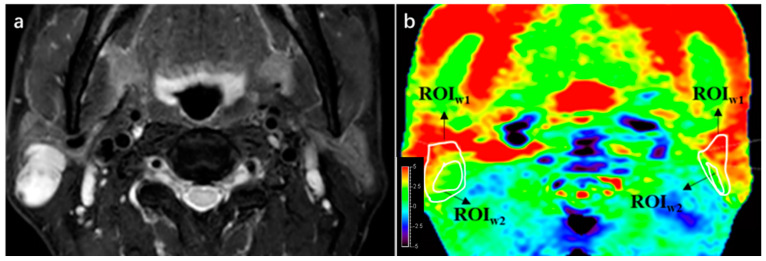
An example of ROI1 and ROI2 drawn in a patient with WT and image quality score. (**a**) A lesion was found at right side of parotid gland in this 55-year-old male on T2WI. (**b**) ROI1 of both the lesion and parotid gland were delineated in APTw sequence 1 imaging according to T2WI, then ROI2 was drawn inside ROI1, maintaining the greatest area of normal gland or lesion while deleting hyperintensity areas. Integrity scores of both lesion and parotid were 4, considering that the entire lesion and parotid gland were displayed. Hyperintensity of lesion and parotid gland were both 3.

**Figure 3 cancers-16-00888-f003:**
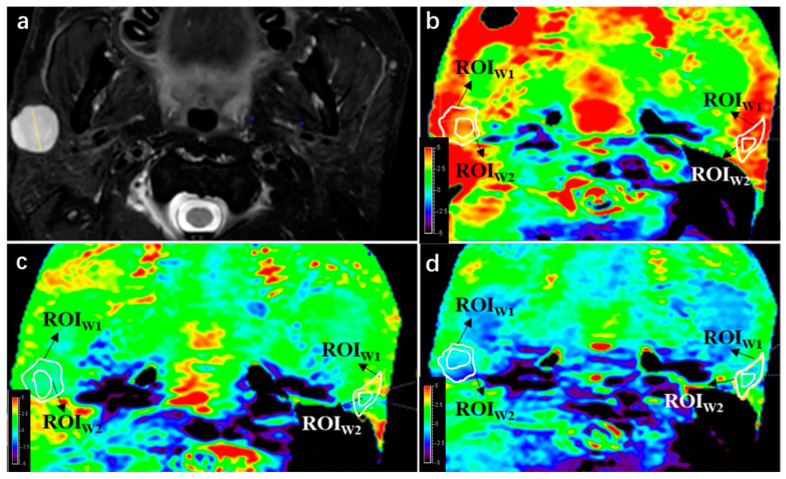
APTw image quality change with B1 power change. (**a**) A lesion was found at right side of parotid gland in this 42-year-old male on T2WI, a pathologically proven pleomorphic adenoma. (**b**) APTw image with B1 = 2 μT; integrity scores of the lesion and parotid gland were both 4, while hyperintensity artifact scores were 3 and 2, respectively. (**c**) APTw image with B1 = 1 μT; integrity scores of the lesion and parotid gland were both 4, respectively, while hyperintensity artifact scores were both 4. (**d**) APTw image with B1 = 0.7 μT; integrity scores of the lesion and parotid gland were 3 and 4, respectively, while hyperintensity artifact scores were both 4.

**Table 1 cancers-16-00888-t001:** Details of MRI Parameters.

Parameters	T1w-mDIXON	T2w-mDIXON	APTw 1	APTw 2	APTw 3
Imaging technique/Orientation	3DTSE/Axial	3DTSE/Axial	3DTSE/Axial	3DTSE/Axial	3DTSE/Axial
TR/TE (msec)	514/7.3	2500/93	6120/8.3	3474/8.3	3474/8.3
Flip angle (degrees)	90	100	90	90	90
FOV (AP × RL × FH) (mm^3^)	200/200/99	200/200/119	230/180/60	230/180/60	230/180/60
Matrix (AP × RL × FH)	252/219/20	31/222/24	128/100/10	128/100/10	128/100/10
Voxel size (mm^3^)	0.8/0.9/4	0.64/0.81/4	1.8/1.8/6	1.8/1.8/6	1.8/1.8/6
Slice gap	1	1	0	0	0
NSA	1.6	2	1	1	1
No. of slices	20	24	10	10	10
TSE factor	8	23	174	174	174
SENSE factor	2.2	2.0	1.6	1.6	1.6
Fat suppression	mDIXON	mDIXON	SPIR	SPIR	SPIR
Saturation B1 rms (μT)	/	/	2	1	0.7
Saturation duration (s)	/	/	2	2	2
Scan time (min/s)	1:45	2:05	3:46	2:09	2:09

**Table 2 cancers-16-00888-t002:** Scores of hyperintensity artifact and their indications.

Scores of Hyperintensity Artifacts	Indication
4	without or with small hyperintensity that does not impair the lesion or parotid gland
3	hyperintensity artifacts impair less than 50% of the lesion or parotid gland
2	hyperintensity artifacts impair more than 50% of the lesion or parotid gland
1	the entire tumor lesion or normal gland is impaired by hyperintensity artifacts

**Table 3 cancers-16-00888-t003:** Characteristics of included patients.

Characteristic	Value
Total number of patients	31
Age (years)	57.74 ± 17.78
Gender (Male/Female)	21/10
Average lesion diameter (mm)	21.29 ± 8.28
Number of patients who received surgery and had pathological diagnosis	22
Age(years)	56.86 ± 18.85
Gender (Male/Female)	17/5
Pathological result	23
Benign tumors	17
Warthin tumor	8
Pleomorphic adenoma	5
Basal cell adenoma	3
Neurilemmoma	1
Malignant tumor	6
Acinar cell carcinoma	2
Lymphoma	2
Adenoid cystic carcinoma	1
Carcinoma ex pleomorphic adenoma	1

3D, three dimensional; APT, amide proton transfer.

**Table 4 cancers-16-00888-t004:** Observer consistence analysis of integrity and hyperintensity artifact score.

Sequence	Score		Kappa Coefficient	95%CI	*p* Value
APT 1	Integrity	PTs (*n* = 32)	0.827	(0.594, 1.060)	<0.001
		PGs (*n* = 30)	0.886	(0.731, 1.041)	<0.001
	Hyperintensity	PTs (*n* = 28)	0.896	(0.757, 1.035)	<0.001
		PGs (*n* = 28)	0.889	(0.754, 1.024)	<0.001
APT 2	Integrity	PTs (*n* = 32)	0.894	(0.749, 1.039)	<0.001
		PGs (*n* = 30)	0.874	(0.705, 1.043)	<0.001
	Hyperintensity	PTs (*n* = 28)	0.781	(0.369, 1.193)	<0.001
		PGs (*n* = 28)	0.885	(0.728, 1.042)	<0.001
APT 3	Integrity	PTs (*n* = 32)	0.913	(0.795, 1.031)	<0.001
		PGs (*n* = 30)	0.884	(0.731, 1.037)	<0.001
	Hyperintensity	PTs (*n* = 28)	0.462	(−0.169, 1.092)	0.015
		PGs (*n* = 28)	0.859	(0.669, 1.049)	<0.001

APT, amide proton transfer; PTs, parotid tumors; PGs, parotid glands; CI, confidence interval.

**Table 5 cancers-16-00888-t005:** Qualitative analysis between parotid lesions and normal parotid glands.

Sequence	Projects		Score 1 (%)	Score 2 (%)	Score 3 (%)	Score 4 (%)	*p* Value
APT 1	Integrity	PTs (*n* = 32)	0	2 (6.25)	6 (18.75)	24 (75.00)	0.082
		PGs (*n* = 30)	2 (6.67)	1 (3.33)	11 (36.67)	16 (53.33)	
	Hyperintensity	PTs (*n* = 28)	9 (32.14)	0	9 (32.14)	10 (35.71)	0.669
		PGs (*n* = 28)	2 (7.14)	9 (32.14)	12 (42.86)	5 (17.86)	
APT 2	Integrity	PTs (*n* = 32)	1 (3.13)	3 (9.38)	12 (37.5)	16 (50.00)	0.723
		PGs (*n* = 30)	0	1 (3.33)	14 (46.67)	15 (50.00)	
	Hyperintensity	PTs (*n* = 28)	0	0	3 (10.71)	25 (89.29)	0.001
		PGs (*n* = 28)	1 (3.57)	3 (10.71)	11 (39.29)	13 (46.43)	
APT 3	Integrity	PTs (*n* = 32)	4 (12.50)	7 (21.88)	13 (40.63)	8 (25.00)	0.022
		PGs (*n* = 30)	0	3 (10.00)	14 (46.67)	13 (43.33)	
	Hyperintensity	PTs (*n* = 28)	0	0	1 (3.57)	27 (96.43)	0.001
		PGs (*n* = 28)	1 (3.57)	1 (3.57)	9 (32.14)	17 (60.71)	

APT, amide proton transfer; PTs, parotid tumors; PGs, parotid glands.

**Table 6 cancers-16-00888-t006:** Qualitative analysis between three APTw sequences.

Sequence	Projects		Score 1 (%)	Score 2 (%)	Score 3 (%)	Score 4 (%)	*p* Value	
PTs	Integrity	APT 1 (*n* = 32)	0	2 (6.25)	6 (18.75)	24 (75.00)	APT 1 vs. APT 2	0.021
		APT 2 (*n* = 32)	1 (3.13)	3 (9.38)	12 (37.5)	16 (50.00)	APT 2 vs. APT 3	0.002
		APT 3 (*n* = 32)	4 (12.50)	7 (21.88)	13 (40.63)	8 (25.00)	APT 1 vs. APT 3	<0.001
	Hyperintensity	APT 1 (*n* = 28)	9 (32.14)	0	9 (32.14)	10 (35.71)	APT 1 vs. APT 2	<0.001
		APT 2 (*n* = 28)	0	0	3 (10.71)	25 (89.29)	APT 2 vs. APT 3	0.157
		APT 3 (*n* = 28)	0	0	1 (3.57)	27 (96.43)	APT 1 vs. APT 3	<0.001
PGs	Integrity	APT 1 (*n* = 30)	2 (6.67)	1 (3.33)	11 (36.67)	16 (53.33)	APT 1 vs. APT 2	0.765
		APT 2 (*n* = 30)	0	1 (3.33)	14 (46.67)	15 (50.00)	APT 2 vs. APT 3	0.206
		APT 3 (*n* = 30)	0	3 (10.00)	14 (46.67)	13 (43.33)	APT 1 vs. APT 3	0.740
	Hyperintensity	APT 1 (*n* = 28)	2 (7.14)	9 (32.14)	12 (42.86)	5 (17.86)	APT 1 vs. APT 2	0.004
		APT 2 (*n* = 28)	1 (3.57)	3 (10.71)	11 (39.29)	13 (46.43)	APT 2 vs. APT 3	0.083
		APT 3 (*n* = 28)	1 (3.57)	1 (3.57)	9 (32.14)	17 (60.71)	APT 1 vs. APT 3	0.003

APT, amide proton transfer; PTs, parotid tumors; PGs, parotid glands.

**Table 7 cancers-16-00888-t007:** Quantitative analysis of parotid lesions and normal parotid glands.

Projects	PTs	PGs	*p* Value
Evaluable group			
Number	28	28	
APT 1 value (%)	3.18 ± 2.74	2.21 ± 2.12	0.144
APT 2 value (%)	0.60 ± 0.75	1.46 ± 1.82	0.024
APT 3 value (%)	0.06 ± 0.83	0.88 ± 1.85	0.036
APT 1–APT 2 (%)	2.59 ± 2.48	0.75 ± 1.20	0.001
APT 1–APT 3 (%)	3.13 ± 2.64	1.33 ± 1.48	0.003
APT 2–APT 3 (%)	0.54 ± 0.83	0.58 ± 0.86	0.860
Trustable group			
Number	19	17	
APT 1 value (%)	1.95 ± 1.77	1.57 ± 2.23	0.570
APT 2 value (%)	0.46 ± 0.67	1.37 ± 2.12	0.085
APT 3 value (%)	0.04 ± 0.84	1.05 ± 2.34	0.086
APT 1–APT 2 (%)	1.49 ± 1.54	0.20 ± 0.51	0.002
APT 1–APT 3 (%)	1.91 ± 1.88	0.51 ± 0.56	0.005
APT 2–APT 3 (%)	0.42 ± 0.90	0.32 ± 0.49	0.666

APT, amide proton transfer; PTs, parotid tumors; PGs, parotid glands.

## Data Availability

Data of this study are unavailable due to privacy and ethical restrictions.

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
