# Peer review of "B1 Power Modification for Amide Proton Transfer Imaging in Parotid Glands: A Strategy for Image Quality Accommodation and Evaluation of Tumor Detection Feasibility"

_cancers, 2024, doi:10.3390/cancers16050888_

Round 1

Reviewer 1 Report

Comments and Suggestions for Authors

In this paper Wu et.al. have shown how moderating B1 power can improve parotid cancer imaging. This is an excellent study but it has few major deficiencies that needs to be improved before acceptance.

Here are the concerns:

a) No background on Amide Proton Transfer, B1 power, T1WI and T2WI were provided. This is not specialized journal on radiology, hence, background information on the basics are required. Very little explanation has been provided and mere citation of papers is not sufficient. After that putting the information and results in perspective is required. 

b) No statistical tests were performed to test the significance of data

c) Figure 2 and 3 have not been explained. What does the color coding mean and how should the audience interpret it?

d) Why or how did they choose B1 saturation? 2, 1 and then it should have been 0.5. Why was 0.7 chosen instead of 0.5?

I look forward to read the revised paper.

Comments on the Quality of English Language

English is fine with minor punctuation errors and typos.

Author Response

Dear reviewer,

            Thank you very much for your comments and professional advice. These opinions help to improve academic rigor of our work. Based on your suggestion and request, we have made corrected modifications on the revised manuscript. Meanwhile, the manuscript had be reviewed and edited for language improvement. We hope that our work can be improved again. Furthermore, we would like to show the details as follows:

  1. a) No background on Amide Proton Transfer, B1 power, T1WI and T2WI were provided. This is not specialized journal on radiology, hence, background information on the basics are required. Very little explanation has been provided and mere citation of papers is not sufficient. After that putting the information and results in perspective is required.

The author’s answer: Thanks for your suggestion. T1WI and T2WI are conventional MRI sequence widely used in clinic but not the main object of our study. B1 and B0 related knowledge had been introduced at the first paragraph of introduction section, and we make it more clear by adding ”(B0)” and “(B1)”to its related contents for better understanding of audiences. The principles and mechanisms of Amide Proton Transfer imaging was introduced in the whole second paragraph of Introduction part. Moreover, we added the relation between APT and B0 and B1 field: ’ The CEST effect is usually small and is sensitive to B0 and B1 fields and saturation pulse power’. As for your suggestion of ‘very little explanation’ and ‘putting the information and results in perspective’, we added some explanation, information and results in perspective at the discussion section. And we would be sincerely grateful to know your further suggestions. Thanks a lot for your review about our work.      

  1. b) No statistical tests were performed to test the significance of data.

The author’s answer: Thanks a lot for your suggestion and we have checked our manuscript several times carefully. However, it seemed that we had performed significance test in every part of statistical analysis of our study, as mentioned in 2.6. Statistical analysis part of the manuscript. Moreover, we have analyzed the statistical tests we chose at the second paragraph of discussion part in the revised edition. We would be very grateful to be noticed where other statistical analysis is needed for our study specifically. 

  1. c) Figure 2 and 3 have not been explained. What does the color coding mean and how should the audience interpret it?

The author’s answer: Thanks a lot for your suggestion. We understand that you might have confusion about the color coding in the color bar provided in Figure 2(b), and we are sorry to find that we had provided a wrong bar in Figures 2, and we have corrected the color bar for color coding (color bar range -5 ~ 5, with percent signs omitted) in Figure 2 and also added the color bar in Figure 3. For better understanding, we also add explanation ‘Area whose APTw value <-5% is defined integrity loss, and whose APTw value > 5% is defined as hyperintensity artifacts, both of them defect imaging quality.’ in 2.4 Qualitative analysis section in the revised edition. Combined with the 4-point Likert scale in Table 2, the interpretation of color coding and score is clearer. We hope these corrections and complements might help you understand our study.

  1. d) Why or how did they choose B1 saturation? 2, 1 and then it should have been 0.5. Why was 0.7 chosen instead of 0.5?

The author’s answer: Thanks for your questions about our study. And the question could be explained partially as we mentioned in the fourth paragraph in 4. Discussion section that ‘CEST imaging is sensitive to B0 and B1 field inhomogeneity, which can induce artifact[33, 34]. In this study, the effect of B0 inhomogeneity to APTw values was minimized by z-spectrum correction based on the B0 field map. Other studies have shown that APTw values increases with B1 power, while CEST contrast is not sensitive to B1 inhomogeneity[35], indicating the potential of using weaker RF to decrease hyperintensity artifacts without attenuating APTw contrast’. Moreover, our previous study had studied the effect of B0 inhomogeneity to the hyperintensity artifacts, so in this study we minimized the B0 inhomogeneity effect, and chose B1 saturation as our study object. Theoretically, APTw value corresponds to particular B1 settings (continuous labeling time and strength), and in this study B1 strength was chosen. As for the third B1 gradient value we chose 0.7 instead of 0.5, is because when B1 = 0.5μT in our pre-experiment, there were large area of signal loss which defected tumor detection. Thanks for your suggestion and we added ‘When B1 = 0.5μT was applied in our pre-experiment, there were large area of signal loss which defected tumor detection, so we chose B1 gradient of 2, 1, 0.7μT’ in the fourth paragraph in the 4. Discussion part for better understanding.

Thank you very much for your attention and time and we are looking forward to hearing from you.

Reviewer 2 Report

Comments and Suggestions for Authors

The manuscript is a study that investigate the effect of B1 power modification for improving Amide Proton Transfer Imaging in Parotid Glands identification. The study is conducted using a cohort of 32 patient and images were evaluated at three different B1 values  

Point 1: I think the summary before the abstract should be moved to the discussion. Moreover, in the abstract please use different font styles for the structured part (Background, Method, results, etc.). Does the w in APTw means “weighted”? What is “ROIw1”, little bit confusion

Point 2: The paper mixes the introduction (problem, motivation, etc.) with the related work (what is in the literature and what is the gap) Please use separate section. Also, the authors need to emphasize the previous work strength and limitations.

Point 3: In Section 2.3, Although a consensus of three readers have been used to for final score determination. Interobserver variability has not been recorded for the study. I am also curious about the effect of the ROI size on the overall results (and thus statistical analysis). Mainly, the second ROI? Please add more details

Point 4: The authors mentioned that one of the limitations is the sample size. I would like to see a power analysis of the appropriate sample size.

Other points

Please define acronyms before their first use, e.g., APTw in the abstract. Also, please review the manuscript language, typos, sentence styles and punctuation, etc. For example “experience respectively” should be “experience, respectively”. There is a space between the sentences and the brackets of the references “previous study[27].”

Comments on the Quality of English Language

Through review of the paper language is highly recommended 

Author Response

Dear reviewer,

            Thank you very much for your comments and professional advice. These opinions help to improve academic rigor of our work. Based on your suggestion and request, we have made corrected modifications on the revised manuscript. Meanwhile, the manuscript had be reviewed and edited for language improvement. We hope that our work can be improved again. Furthermore, we would like to show the details as follows:

  1. I think the summary before the abstract should be moved to the discussion. Moreover, in the abstract please use different font styles for the structured part (Background, Method, results, etc.). Does the w in APTw means “weighted”? What is “ROIw1”, little bit confusion

The author’s answer: Thanks for your advices. 1) The simple summary part before abstract is a compulsory manuscript component of the journal cancers. Therefore, we are sorry that we could not be able to delete the summary, but we put the summary content into the discussion as you suggested for better understanding of our work. 2) The structured part (Background, Methods, Results and Conclusions) is bolded in the revised edition. 3) The w in APTw means ‘weighted’, and we have deleted the w in ROIw1 and ROIw2 in the revised edition to avoid the confusion.    

  1. The paper mixes the introduction (problem, motivation, etc.) with the related work (what is in the literature and what is the gap) Please use separate section. Also, the authors need to emphasize the previous work strength and limitations.

The author’s answer: Thanks for your suggestion. 1) In the introduction section, the first paragraph introduces diagnosis of parotid tumors nowadays; the second paragraph introduces APT and its related principles; the third and fourth paragraph are about the studies of application of APT and the hindrance in head and neck area, which are the problems and also the motivation of our work because the APT is still on research stage. Since APT is a technique which is not applied clinically, we assume the problem, motivation and the related literature work might not be separated. And we would be sincerely grateful to know your different opinions. 2) Strength and limitations of previous work are emphasized on the third paragraph of discussion section in the revised edition.     

  1. In Section 2.3, Although a consensus of three readers have been used to for final score determination. Interobserver variability has not been recorded for the study. I am also curious about the effect of the ROI size on the overall results (and thus statistical analysis). Mainly, the second ROI? Please add more details

The author’s answer: Thanks for the suggestions. 1) Interobserver variability analysis had been introduced in 2.4 section that “The two radiologists independently scored the integrity and hyperintensity artifact of PTs and PGs, and the consistency between their assessments was analyzed. In case where there were initially inconsistent scores, a third senior radiologist reviewed the images and made the final decision”, which was bolded in the revised edition. The description of related analysis methods and results were bolded at section 2.6 and 3.2 (table 4) as well. 2)We are sincerely sorry that we could not be able to use the ROI data in this study to address the effect of the ROI size on the overall results, because the ROI size had not been measured and recorded in this study since our initial target is to evaluate the integrity and hyperintensity of ROI1, and APTmean value of ROI2. We would like to measure the size in the revision process if the former ROI contour was saved, but the software to draw the ROI did not have the storage function and the ROIs we drew again would be different from the former ones. ROI might have effect on overall results indeed, but our methods of drawing ROI1(delineating the entire parotid glands or tumors based on the T2W images) and ROI2 (to maintain the most parotid gland or lesion area while exclude most of the hyperintensity artifacts) have tried our best to decrease the effect. And the ROIs were drawn by two radiologists independently, the consistency of integrity and hyperintensity score might help prove the little effect of ROI size.       

  1. Point 4: The authors mentioned that one of the limitations is the sample size. I would like to see a power analysis of the appropriate sample size.

The author’s answer: Thanks for your suggestion. According to our pre-experiment published (Yu Chen et al., 2021), the difference between parotid tumors and parotid glands is significant in trustable group (1.99%±1.18% vs. 1.03%±1.09%, p=0.018). When α=0.05, β=0.2, μ1, μ2, σ1 and σ2 are set as listed above, N1 and N2 are calculated as 23 and 23 to get a significant result, respectively (PASS 2021, v21.0.3). Therefore, 23 parotid tumors and 23 parotid glands in trustable group are required to get a significant difference. In this study after image quality evaluation and selection, only 19 parotid tumors and 17 parotid glands were left in the trustable group in the analysis. Related analysis has been added to the Discussion section.  

  1. Please define acronyms before their first use, e.g., APTw in the abstract. Also, please review the manuscript language, typos, sentence styles and punctuation, etc. For example “experience respectively” should be “experience, respectively”. There is a space between the sentences and the brackets of the references “previous study[27].”

The author’s answer: Thanks a lot for your suggestion. Acronyms and language mistakes including these you pointed out have been corrected in the revised edition.

  1. Through review of the paper language is highly recommended.

The author’s answer: Thanks for your advice, we have checked our manuscript by a colleague fluent in English writing.

Thank you very much for your attention and time and we are looking forward to hearing from you.

Reviewer 3 Report

Comments and Suggestions for Authors

In the manuscript entitled “B1 Power Modification for Amide Proton Transfer Imaging in Parotid Glands: A Strategy for Image Quality Accommodation and Evaluation of Tumor Detection Feasibility”, Wu et al. present insights into the application of amide proton transfer (APTw) imaging in the context of parotid tumors, focusing on the impact of B1 power modification on image quality and tumor detection. However, there are several areas where significant revisions are warranted to enhance the clarity, depth, and academic rigor of the paper.

1.     The manuscript should provide a more detailed contextualization within the existing literature on APTw imaging, particularly in relation to parotid tumors. This could include a comparison of how this study's findings either align with or diverge from previous research, offering a clearer contribution to the field.

2.     The methodology section would benefit from a more comprehensive justification for the inclusion and exclusion criteria, especially the specific age range and the rationale behind excluding patients with certain artifacts. Additionally, more detailed information about the MRI techniques, particularly the specifics of the 3D TSE APTw MRI, would enhance the reproducibility and robustness of the study.

3.     In terms of data analysis, while the use of a 4-point Likert scale and the delineation of regions of interest (ROI) are commendable, the paper could greatly benefit from a more in-depth statistical analysis. This includes a thorough discussion on the choice of statistical tests used and their suitability for the type of data and research questions posed.

4.     The results section, though detailed, requires a deeper exploration of the implications of the findings. For example, an analysis of why significant differences in APTmean values were observed between parotid tumors and glands in some sequences but not others would provide greater insight and contribute to a more nuanced understanding of the study's outcomes.

5.     The discussion of limitations and future research is well-acknowledged; however, it could be expanded. Specific suggestions for future research methodologies or study designs that could address these limitations would be beneficial. This would not only strengthen the current study but also guide subsequent research in the field.

In conclusion, while the manuscript provides important contributions to the understanding of APTw imaging in parotid tumors, enhancements in literature contextualization, methodology clarity, depth of statistical analysis, interpretation of results, and future research directions are needed to elevate the academic and practical value of the work.

Author Response

Dear reviewer,

            Thank you very much for your comments and professional advice. These opinions help to improve academic rigor of our work. Based on your suggestion and request, we have made corrected modifications on the revised manuscript. Meanwhile, the manuscript had be reviewed and edited for language improvement. We hope that our work can be improved again. Furthermore, we would like to show the details as follows:

  1. The manuscript should provide a more detailed contextualization within the existing literature on APTw imaging, particularly in relation to parotid tumors. This could include a comparison of how this study's findings either align with or diverge from previous research, offering a clearer contribution to the field.

The author’s answer: Thanks for your advice, and a more detailed contextualization within the existing literature on APTw imaging, particularly in relation to parotid tumors have been added as the third paragraph of discussion section.

  1. The methodology section would benefit from a more comprehensive justification for the inclusion and exclusion criteria, especially the specific age range and the rationale behind excluding patients with certain artifacts. Additionally, more detailed information about the MRI techniques, particularly the specifics of the 3D TSE APTw MRI, would enhance the reproducibility and robustness of the study.

The author’s answer: Thanks for your advice. 1)The age range was not limited in this study so we have deleted it from the inclusion criteria. We excluded the four patients as the exclusion criteria is because images of the four patients were too severely damaged to be enrolled into further analysis, and we added related description there. 2) 3D TSE APTw MRI is widely recommended in brain tumors, and we added more detailed introduce of it at introduction section: “Based on previous studies of APTw imaging on brain tumor, a fast 3D acquisition technique, integrated with a feasible, optimized RF saturation scheme and an effective lipid suppression method is recommended and TSE has been recommended among the candidate readout sequences[25]”.    

  1. In terms of data analysis, while the use of a 4-point Likert scale and the delineation of regions of interest (ROI) are commendable, the paper could greatly benefit from a more in-depth statistical analysis. This includes a thorough discussion on the choice of statistical tests used and their suitability for the type of data and research questions posed.

The author’s answer: Thanks for your advice, and the thorough discussion is added as the second paragraph of discussion part.

  1. The results section, though detailed, requires a deeper exploration of the implications of the findings. For example, an analysis of why significant differences in APTmean values were observed between parotid tumors and glands in some sequences but not others would provide greater insight and contribute to a more nuanced understanding of the study's outcomes.

The author’s answer: Thanks for your advice. And an analysis of the result observed has been added into the fifth paragraph of discussion section.

  1. The discussion of limitations and future research is well-acknowledged; however, it could be expanded. Specific suggestions for future research methodologies or study designs that could address these limitations would be beneficial. This would not only strengthen the current study but also guide subsequent research in the field.

The author’s answer: Thanks for your suggestion, and the specific suggestion for future research has been added at the last paragraph of discussion section in revised edition.

Thank you very much for your attention and time and we are looking forward to hearing from you.

Round 2

Reviewer 1 Report

Comments and Suggestions for Authors

The authors have significantly improved the manuscript especially the discussion. The work now makes more sense and significant.

Congratulations!

Reviewer 2 Report

Comments and Suggestions for Authors

The authors addressed the comments in the previous round. 

Comments on the Quality of English Language

NA

Reviewer 3 Report

Comments and Suggestions for Authors

The authors have addressed all my comments in their revision. I can now recommend the publication of this manuscript.